# Atmospheric Torques and Earth's Rotation: What Drove the Millisecond-Level Length-of-Day Response to the 2015-16 El Niño?

Sébastien B. Lambert [1], Steven L. Marcus [2], and Olivier de Viron [3]

[1]SYRTE, Observatoire de Paris, PSL Research University, CNRS, Sorbonne Universités, UPMC Univ. Paris 06, LNE, Paris, France
[2]Independent Researcher, Santa Monica, California, USA
[3]Littoral, Environnement et Sociétés (LIENSs), Université de La Rochelle and CNRS (UMR7266), La Rochelle, France

*Correspondence to:* Sébastien B. Lambert (sebastien.lambert@obspm.fr)

**Abstract.** El Niño/Southern Oscillation (ENSO) events are classically associated with a significant increase in the length of day (LOD), with positive mountain torques arising from an east-west pressure dipole in the Pacific driving a rise of atmospheric angular momentum (AAM) and consequent slowing of the Earth's rotation. The large 1982-83 event produced a lengthening of the day of about 0.9 ms, while a major ENSO event during the 2015-16 winter season produced an LOD excursion reaching 0.81 ms in January 2016. By evaluating the anomaly in mountain and friction torques, we found that: (i) as a mixed Eastern/Central Pacific event, the 2015-16 mountain torque was smaller than for the 1982-83 and 1997-98 events which were pure Eastern Pacific events, and (ii) the smaller mountain torque was compensated by positive friction torques arising from an enhanced Hadley-type circulation in the Eastern Pacific, leading to similar AAM/LOD signatures for all three extreme ENSO events. The 2015-16 event thus contradicts the existing paradigm that mountain torques cause the Earth rotation response for extreme El Niño events.

## 1 Introduction

Earth rotation fluctuates with time, as a response to the interaction of the solid Earth with celestial bodies, the liquid core, and the fluid layers of the climate system. This interaction results in changes of the orientation of the Earth rotation vector in space, of the orientation of the Earth around its rotation axis, and of the Earth rotation angular velocity associated with changes in the length of the day (LOD). Variations of the LOD can reach a few milliseconds on the timescale of a few tens of years, due to core-mantle interaction (Holme and de Viron, 2013, and references therein), and a few tenths of a millisecond on the timescale of some days to several years, due mostly to solid-Earth/atmosphere interaction (Hide and Dickey, 1991), though the solid-Earth/ocean interaction does play a small role (Marcus et al., 1998; Dickey et al., 2010; Marcus et al., 2012). A major atmospheric impact on Earth rotation occurs on the annual timescale (Hide et al., 1998), due to the hemispheric asymmetry of the seasonal cycle (de Viron et al., 2002), with El Niño/Southern Oscillation (ENSO) events (Chao, 1984; Carter et al., 1984) dominating interannual (2-7 year) variability.

Extreme El Niño events, such as those that occurred in the winters of 1982-83, 1997-98 and 2015-16 (e.g., http://ggweather.com/enso/oni.htm), can generate LOD anomalies reaching amplitudes of nearly a millisecond with respect to

the climatological seasonal cycle. Previous studies found that the creation of the large LOD anomaly during the 1982-83 event was mainly due to mountain torque on the American and Eurasian orography (Ponte et al., 1994; Ponte and Rosen, 1999). In general, the paradigm has emerged that mountain torques (defined more precisely below) generate the rotational anomalies associated with the El Niño cycle, while friction torques play a more passive role by damping these anomalies back towards their climatological norms.

Considering two types of El Niño events defined in the recent literature (see below), de Viron and Dickey (2014) showed that Central Pacific (CP) events are associated with smaller LOD anomalies than Eastern Pacific (EP) events, due to the position and amplitude of the pressure anomaly over the Pacific ocean that generates a weaker mountain torque. While 1982-83 and 1997-98 are cited as examples of classical EP events, Paek et al. (2017) noted the different nature of the extreme 2015-16 episode, finding it to be the strongest mixed EP/CP event ever recorded. In this paper we seek to document and understand how the different atmospheric torques active during the recent mixed event raised the atmospheric angular momentum (AAM) and consequently the LOD anomalies to values similar to those reached during the previous extreme EP events. As an extreme event of a unique nature, the 2015-16 mixed EP/CP episode offers a chance to gain further insights into how atmospheric dynamics link Earth rotation anomalies to different 'flavors' of El Niño (Johnson, 2013).

## 2   Methods and Data

When studying the impact of the atmosphere on Earth rotation, two different approaches can be used. First, one can consider that the atmosphere is included in the Earth system, compute the variation of the AAM, consider that the angular momentum of the system is conserved, so that what is lost by the atmosphere is gained by the solid Earth, and estimate from there the Earth rotation change: this is known as the angular momentum approach. The other approach considers the atmosphere as an external forcing on the solid Earth, computes the torque exerted by the atmosphere on the solid Earth, and estimates the Earth rotation changes by the angular momentum budget equation. Note that the angular momentum approach (applied below in Fig. 2) is preferred for explicating (or predicting) LOD anomalies, which can serve as external 'ground truth' for validating model-specified (or predicted) AAM under the severe conditions associated with extreme ENSO episodes. On the other hand, the torque approach (applied below in Fig. 4) can provide dynamical insight into the mechanisms generating the near-millisecond LOD anomalies which accompany these events, and can also be used for internal consistency checks of the model AAM budgets under the strong perturbations involved.

As shown in Barnes et al. (1983), the total torque exerted by the atmosphere on the Earth is composed of three effects: the gravitational attraction of the mass anomalies inside the Earth by those inside the atmosphere, the atmospheric pressure acting over the topography, and the friction of the wind on the surface. The first two contributions are classically merged into the so-called *mountain* torque while the latter is known as *friction* torque.

We used standard formulations of the AAM, and mountain and friction torques that can be found in, e.g., Huang et al. (1999). The AAM is composed of two parts, a *mass* term corresponding to the angular momentum associated with the rigid rotation of the atmosphere with the solid Earth and a *motion* term corresponding to the relative angular momentum of the atmosphere

with respect to the solid Earth. The $Z$ component of the AAM was estimated from

$$H_Z^{\text{mass}} \quad = \quad \frac{a^4\Omega}{g} \int\limits_0^{2\pi} \int\limits_0^{\pi} P_{\text{s}} \sin^3\theta \mathrm{d}\theta \mathrm{d}\lambda, \tag{1}$$

$$H_Z^{\text{motion}} \quad = \quad \frac{a^3}{g} \int\limits_0^{2\pi} \int\limits_0^{\pi} \int\limits_0^{P_{\text{s}}} u \sin^2\theta \mathrm{d}p \mathrm{d}\theta \mathrm{d}\lambda, \tag{2}$$

where $a$ is the mean Earth radius, $g$ is the mean gravity acceleration, $u$ is the zonal wind, $P_{\text{s}}$ is the surface pressure, $\theta$ and $\lambda$ are the colatitude and longitude, respectively, and $\Omega$ is the Earth mean angular velocity. The axial torques were estimated using

$$\Gamma_Z^{\text{mountain}} \quad = \quad a^3 \int\limits_0^{2\pi} \int\limits_0^{\pi} \frac{\partial P_{\text{s}}}{\partial \lambda} h \sin\theta \mathrm{d}\theta \mathrm{d}\lambda, \tag{3}$$

$$\Gamma_Z^{\text{friction}} \quad = \quad -a^3 \int\limits_0^{2\pi} \int\limits_0^{\pi} \tau_\lambda \sin^2\theta \mathrm{d}\theta \mathrm{d}\lambda, \tag{4}$$

where $h$ is the orography and $\tau_\lambda$ is the zonal friction drag. The time rate of change of the total AAM is given by the sum of the mountain and friction torques (e.g., Barnes et al., 1983):

$$\frac{\mathrm{d}}{\mathrm{d}t} \left( H_Z^{\text{mass}} + H_Z^{\text{motion}} \right) = \Gamma_Z^{\text{mountain}} + \Gamma_Z^{\text{friction}}. \tag{5}$$

In what follows, the time-integrated form of Equation (5) was used to evaluate the sources of AAM maxima associated with recent extreme El Niño events. Given the AAM variation, the induced change in the LOD is estimated by

$$\frac{\Delta\text{LOD}}{\overline{\text{LOD}}} = \frac{\Delta\left( 0.7 H_Z^{\text{mass}} + H_Z^{\text{motion}} \right)}{C\Omega}, \tag{6}$$

where $\overline{\text{LOD}}$ is the nominal length of the Solar day (86400 s) and $C$ is the axial mean moment of inertia of the Earth; **the mass term is evaluated using the inverted barometer assumption (Jeffreys, 1916) to account for the quasi-static response of the oceans to atmospheric pressure loading, and the factor of 0.7 accounts for the compensating changes in the moment of inertia arising from the elastic deformation of the solid Earth in response to the surface loading (Munk and MacDonald, 1960; Barnes et al., 1983).**

Our computations of AAM and torques were based on $2° \times 2°$ surface pressure, zonal momentum flux, and zonal wind speed data from daily and monthly values from the European Center for Medium-range Weather Forecast (ECMWF) ERA Interim model Dee et al. (2011) spanning 1979-2017. Wind speeds were taken at 17 pressure levels between 10 and 1000 hPa. The longitudinal gradients of the pressure field were computed with a five-point stencil.

For computation of the mountain torque, we used the model orography at its native $2° \times 2°$ resolution, thereby ensuring consistency between the wind, pressure, and zonal momentum flux data sets and the derived AAM and torque quantities. A recent study by van Niekerk et al. (2016) found that resolved mountain torques in the Met Office United Model with free atmospheric wind and temperature relaxed to ERA Interim reanalyses are relatively insensitive to increasing model resolution

(see, e.g., their Fig. 7), although they are more strongly impacted by large-scale (synoptic) processes than are the parameterized sub-grid scale torques (not considered in our study).

Earth rotation data were provided by the International Earth rotation and Reference systems Service (IERS) Earth Orientation Parameter (EOP) 14 C 04 series available via the IERS Earth Orientation Center Web site (http://iers.obspm.fr/eop-pc). This combination of very long baseline interferometry (VLBI), global navigation satellite systems (GNSS), Doppler orbitography (DORIS), and satellite-laser ranging (SLR) data provides daily estimates of the LOD with an accuracy of about 0.05 ms. To isolate as much as possible the anomalous changes due to episodic events like ENSO, we subtracted modeled zonal tides (Petit and Luzum, 2010), a multidecadal trend estimated with a 4-yr running mean and a 5.9-yr periodic term, attributed to secular tidal braking / post-glacial rebound (Hide and Dickey, 1991) and variations of the fluid core angular momentum (Hide et al., 2000; Holme and de Viron, 2013), and a mean seasonal cycle estimated over 1979-2017. The residual LOD contains essentially the fluctuation associated with anomalous AAM and oceanic currents, with the latter being less than 5%.

As measurements of ENSO activity, we used monthly series of Niño 1+2, Niño 3, Niño 4, and Niño 3.4 indices retrieved from the Climate Prediction Center (CPC) of NOAA. We used the Niño 1+2, Niño 3, and Niño 4 series to compute the indices relevant to Eastern Pacific (EP) and Central Pacific (CP) events considered in (Takahashi et al., 2011, their formulae 3 and 4) and (Ren and Jin, 2011, their formula 1). For comparison, we also used corresponding EP/CP indices provided by Kao and Yu (2009) at https://www.ess.uci.edu/ yu/2OSC and the PT' indices of (Chen and Wallace, 2016, private communication).

We removed seasonal composites and linear trends from all grids and time series and we formed two-month 'winter' values of all the above time-variable quantities preceding the respective AAM/LOD maxima, by averaging over December-January for the EP events, and over November-December for the mixed EP/CP event (see below). For computation of regional torques, we used a modified version of the land-sea masks whose geographical limits can be seen in Fig. 3 of Marcus et al. (2011) with limits of the equatorial zone set to $\pm 15°$ from the equator. Moreover we separated Greenland from North America and the Pacific ocean into Eastern and Western zones, respectively east and west from the International Date Line.

## 3 Analysis and Results

For the three extreme El Niño events that occurred in the last forty years (winters of 1982-83, 1997-98, and 2015-16) the Niño 3.4 index reached values of more than two times its standard deviation over the winter season. A feature that makes the 2015-16 ENSO event unique is the hybrid aspect mentioned by Paek et al. (2017) who showed that EP/CP indices defined by Kao and Yu (2009) were of comparable magnitudes, in contrast to their highly positive EP index and small or negative CP index in 1997-98. Similar conclusions can be drawn on the basis of projections provided by Takahashi et al. (2011) and Ren and Jin (2011), the former seeing the 2015-16 event as more of a CP and the latter as more of an EP type (Fig. 1). Chen and Wallace (2016), who considered a methodology similar to Takahashi et al. (2011) but considering also extra-tropical Pacific variability, derived indices of equal magnitudes by including contributions of higher latitude SST.

For the three events, anomalous LOD excursions reached the level of nearly a millisecond, consistent with the AAM variation (Fig. 2). The Niño 3.4 index reached its maximum values between one and three months before AAM and LOD (Dickey et al.,

2007). Differences between AAM and LOD anomalies might be partly due to a small, variable contribution from the ocean and hydrology and to local biases or side effects induced by the filtering/smoothing method used to separate the non-seasonal LOD from its multidecadal and interannual trends. We found that the LOD for the 2015-16 event peaked at 0.81 ms on 6 January 2016. Figure 2 suggests also that the 1982-83 event was the strongest from the Earth rotation point of view, generating an

LOD anomaly of 0.91 ms, that is about 3.5 times the standard deviation of the mean seasonal cycle. The 1997-98 event was somewhat less active with an LOD anomaly of only 0.76 ms. Our values are consistent with analyses of Gipson (2016) based on very long baseline interferometry data, who found comparable excursions of the LOD in 1997-98 and 2015-16 of about 0.75 ms; interestingly, however, the maximum rotational anomalies for the two earlier EP events occur nearly a month later in the season than for the 2015 mixed event.

In order to analyze the synoptic features giving rise to these rotational anomalies, we formed global maps of the surface pressure and surface friction drag anomalies for the two months which preceded them, averaging over Dec-Jan for the 1982-83 and 1997-98 winters, and over Nov-Dec for the 2015-16 winter (Fig. 3). The two EP surface pressure maps (1982-83 and 1997-98) reveal the classic east-west dipole for this type of event noted by de Viron and Dickey (2014), with the low pressure areas in proximity to the American coast generating substantial positive mountain torques on the atmosphere and thereby increasing the

LOD. For the mixed EP/CP event in 2015, however, the surface pressure gradients have a substantial meridional component, with a low pressure area in the equatorial east-central Pacific flanked by anomalous high pressure zones in the northeastern (NE) and southeastern (SE) Pacific. This Hadley-type pattern gives rise to anomalous easterlies in the eastern equatorial (EE) Pacific, generated as inflow to the equatorial low-pressure area near 120°W, and also in the NE and SE Pacific, generated as enhanced easterly flow on the equatorward flanks of the anomalous Pacific high pressure areas. The result is a significant enhancement of

positive friction torque over the eastern tropical and mid-latitude Pacific, denoted by the orange-shaded areas in the right-hand column of Fig. 3, for the mixed EP/CP event as compared with the earlier EP events. This comparison is highlighted in the bottom row of Fig. 3, which shows the difference of the surface pressure and friction drag anomalies between the two-month means for the mixed event and the average of the two EP events. The pressure difference (left panel) shows that the change between the mixed and EP events takes the form of a strengthened Hadley-type circulation in the east-central Pacific, with

the stronger and more equatorward response in the winter (northern) hemisphere. This is reflected in the friction difference between the mixed and EP events (right panel), which shows a strong enhancement of the surface drag in the NE Pacific and the EE Pacific; a weaker enhancement is also seen over the mid-latitude SE Pacific, compensated by enhanced westerlies over the Antarctic Circumpolar Current.

The main features observed in the maps are reflected by the values of the two-month averaged mountain and friction torque

anomalies for the various land and ocean areas given in Table 1. The lower net 2015-16 mountain torque compared to 1982-83 and 1997-98 is consistent with the lack of a pronounced east-west pressure dipole in the Pacific in late 2015, resulting in a substantial negative torque over North America / Greenland during that time, while the low pressure anomaly in the equatorial Pacific generated similar positive mountain torques over South America as for the EP events. A significant portion of the remaining difference in global mountain torque between the three events is generated over the European continent, as a

consequence of changes in the relative positions of large-scale North Atlantic features during the winter, possibly influenced

by El Niño (Butler et al., 2014), that modify the direction and the intensity of the downstream pressure gradients over the continental orography (particularly the Caucasus and Zagros Mountains). The lack of a net positive mountain torque leading up to the 2015-16 rotational maximum, however, is compensated by the presence of positive friction torques during that time, particularly over the eastern Pacific, with the NE and EE regions making the largest contributions, relative to their corresponding values during the EP events; the intense high pressure area in the SE Pacific, reminiscent of the November 2009 feature discussed by Lee et al. (2010), makes a smaller positive contribution to the CP-EP axial torque difference due to its higher latitude.

The contributions of these processes to the LOD maxima generated during the three extreme events can be illustrated by integrating the daily torques over preceding intervals in the time domain to reconstitute the AAM. The difference between the integrated friction and mountain torques and the AAM arising from gravity-wave drag and other torques related to the sub-grid scale orography is generally considered to be negligible at these time scales (de Viron et al., 1999; Ponte and Rosen, 1999). As starting epochs, we chose the beginning of the rise of each AAM curve towards its peak value. The resulting reconstituted AAM components - consistent with Figs. 2 and 3 of Ponte and Rosen (1999) for the 1982-83 event - are shown in Fig. 4. They reveal that a positive friction episode occurred in 2015-16 about 15 to 20 days before the AAM peak. Such a positive friction episode, occurring about 10 to 15 days before the AAM peak, was totally absent from the 1997-98 event and was much smaller in the 1982-83 event, when the integrated friction torque remains positive during a few days before turning back to negative values. Note that choosing starting epochs and integration period consistent with the two-month intervals considered above leads to similar conclusions but to less consistent closures of the budget due to small biases accumulated in the integration, as already mentioned in Ponte and Rosen (1999). The lower panel of Fig. 4 demonstrates the importance of the Eastern Pacific contribution to the overall positive friction torque in the last two weeks of 2015, and highlights absolute contributions from the EE and SE Pacific regions at this time. The NE Pacific also contributes positively, but to a lesser extent; its contribution relative to the corresponding (negative) NE Pacific values for the EP events, however, is greater than those for the EE and SE regions combined over the two months preceding these events (Table 1). A Hovmoeller (time-latitude) plot of the Eastern Pacific frictional drag contributing to the 2015-16 LOD maximum (Fig. 5) highlights its three-belt structure, and shows the EE Pacific contribution (spanning 15°N-15°S) to arise from two areas: one in the southern hemisphere originating from inflow to the westward-displaced boreal winter Hadley circulation, and one in the northern hemisphere originating from enhanced easterly flow on the equatorward flank of the NE Pacific high pressure area (similar to the Nov-Dec 2015 pressure anomalies seen in Fig. 3e).

## 4  Discussion and Conclusion

Surface pressure and friction torque anomaly maps for the last two months before each extreme AAM/LOD peak (Fig. 3) and a time-latitude plot of frictional stress during the last event (Fig. 5) suggest that the 2015-16 positive friction torque arises from three zones: in the NE Pacific between latitudes of 0 and 40°N, in the EE Pacific off Peru, and in the SE Pacific between 40°S and 60°S, showing rough symmetry about an enhanced (boreal winter) Hadley-type circulation in the East-Central Pacific. The

positive (LOD-lengthening) EE Pacific contribution to the friction torque can be understood in the context of a CP event in which the ENSO-driven convection is displaced towards the central Pacific; the incoming winds that supply the convection are westward in the EE Pacific. From the momentum point of view, this convection also strengthens the Hadley circulation and the subtropical jets that carry the bulk of the AAM signal. The SE Pacific positive torque contribution results from a strengthening of the extratropical South Pacific anticyclone similar to that documented in November 2009 during a CP event (Lee et al., 2010), with the NE Pacific positive contribution stemming from a similar high pressure response in the subtropical winter hemisphere. Note, however, that the weaker 2009-10 CP event, which lacked the NE Pacific circulation center found during the extreme 2015-16 episode, did not produce a significant anomaly in AAM or LOD.

The three extreme ENSO events of 1982-83, 1997-98, and 2015-16 were of comparable strengths expressed through both SST and subsurface indices (L'Heureux et al., 2017). The latest was, however, of a different nature, as a mixed EP/CP event, as opposed to the other two pure EP events (Paek et al., 2017; Palmeiro et al., 2017). All three events produced anomalous excursions of the Earth's LOD between 0.76 ms (1997-98) and 0.91 ms (1982-83), the amplitude of the 2015-16 (0.81 ms) excursion being intermediate. We showed that, though the 1982-83 and 1997-98 LOD anomalies were driven by the mountain torque, as expected with pure EP events, the LOD excitation mechanism of the mixed EP/CP 2015-16 event was different. The weaker mountain torque was compensated by a positive friction torque acting in the Eastern Pacific, both in an absolute sense and relative to the frictional torques prevailing there during the earlier EP events. The 2015-16 event, unique for its nature and intensity among the ENSO events recorded for the last four decades, thus contradicts the existing paradigm that mountain torques cause the Earth rotation response for extreme El Niño events.

For mixed or CP events, increasing in frequency and strength since the turn of the century (Wang and Cai, 2013), friction torques arising form tropical and extratropical centers of action can make a significant contribution to the positive LOD anomalies, thereby compensating for the less efficient CP mountain torque coupling (de Viron and Dickey, 2014) and maintaining the capability for a robust rotational response to this new 'flavor' of event. Interestingly, enhanced easterlies (or positive friction; England et al., 2014), more and stronger CP events (Lee and McPhaden, 2010; Paek et al., 2017), and the global warming hiatus (Douville et al., 2015; Song et al., 2016) have coincided in the early 21st century; the chain of causality among these events, however, is far from clear.

These three extreme events also exemplify the complex relation between the ENSO strength and the atmospheric response (AAM and torques) that leads to variations in the Earth's rotation rate. The dominant factor is the position and the depth of the ENSO pressure dipole that can significantly strengthen or weaken the mountain torques exerted by the atmosphere on the Andes and the Rockies. Nevertheless, the factors governing teleconnections between tropical Pacific SST anomalies and the globally-distributed pressure/wind response, which are still being actively investigated (e.g., Ji et al., 2016) may play a critical role in determining the relative rotational signatures of the events.

*Author contributions.* SL proposed the idea for this study. All authors contributed to develop the methods, analyzed the data and contributed the materials/analysis tools. All the authors wrote jointly the paper and discussed extensively the results and the interpretations. All authors read and approved the final manuscript.

*Competing interests.* The authors declare that they have no competing interests.

5  *Copyright statement.* All the authors have agreed to the licence and copyright agreement.

*Acknowledgements.* The work of OdV was financially supported by CNES, through the TOSCA program, as an application of the geodesy missions.

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

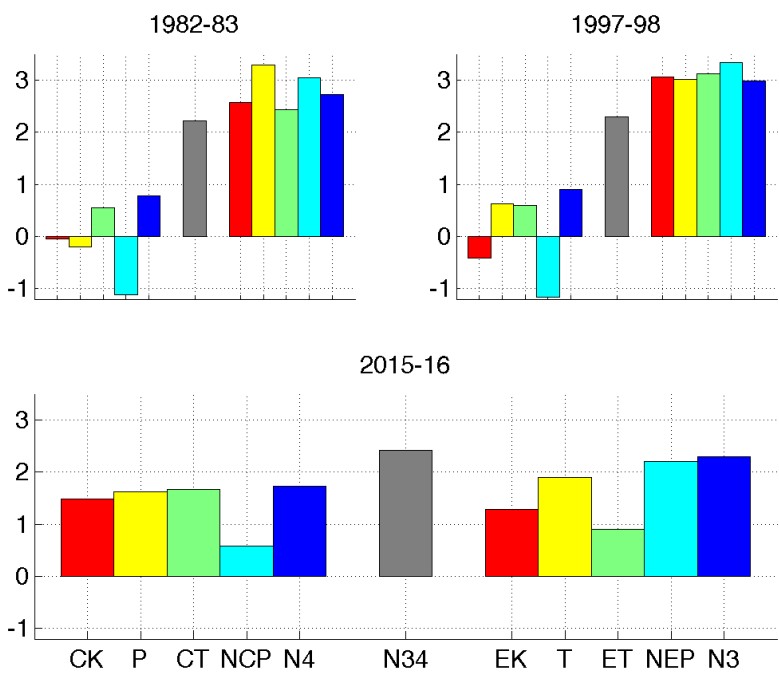

**Figure 1.** The DJF Niño 3.4 (N34), Niño 3 (N3), and Niño 4 (N4) indices and other various 'Eastern' and 'Central' Pacific projection indices for each of the three events. Eastern (Central) Pacific indices are represented right (left) of the Niño 3.4 bar. EK/CK: E and C indices from Kao and Yu (2009); P/T: P and T' indices from Chen and Wallace (2016); ET/CT: E and C indices from Takahashi et al. (2011); NEP/NCP: EP (also CT) and CP (also WP) indices from Ren and Jin (2011).

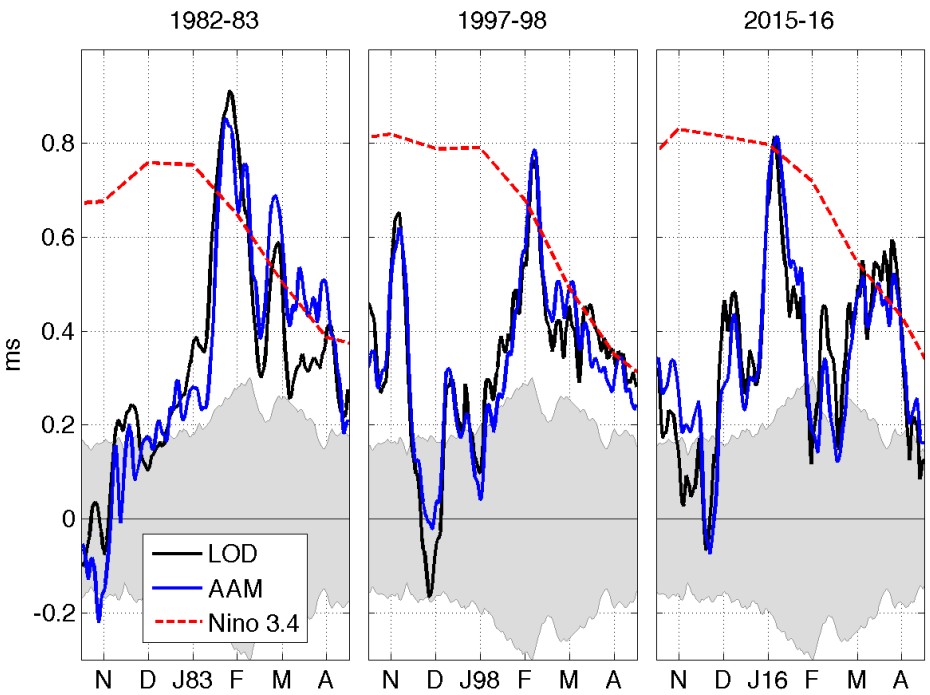

**Figure 2.** The time series of daily AAM (blue) and LOD (black) values around the three extreme events. The red, dashed line represents the scaled monthly Niño 3.4 index. The shaded area represents one standard deviation around the climatological mean. The x-axis ticks indicate the first day of each month.

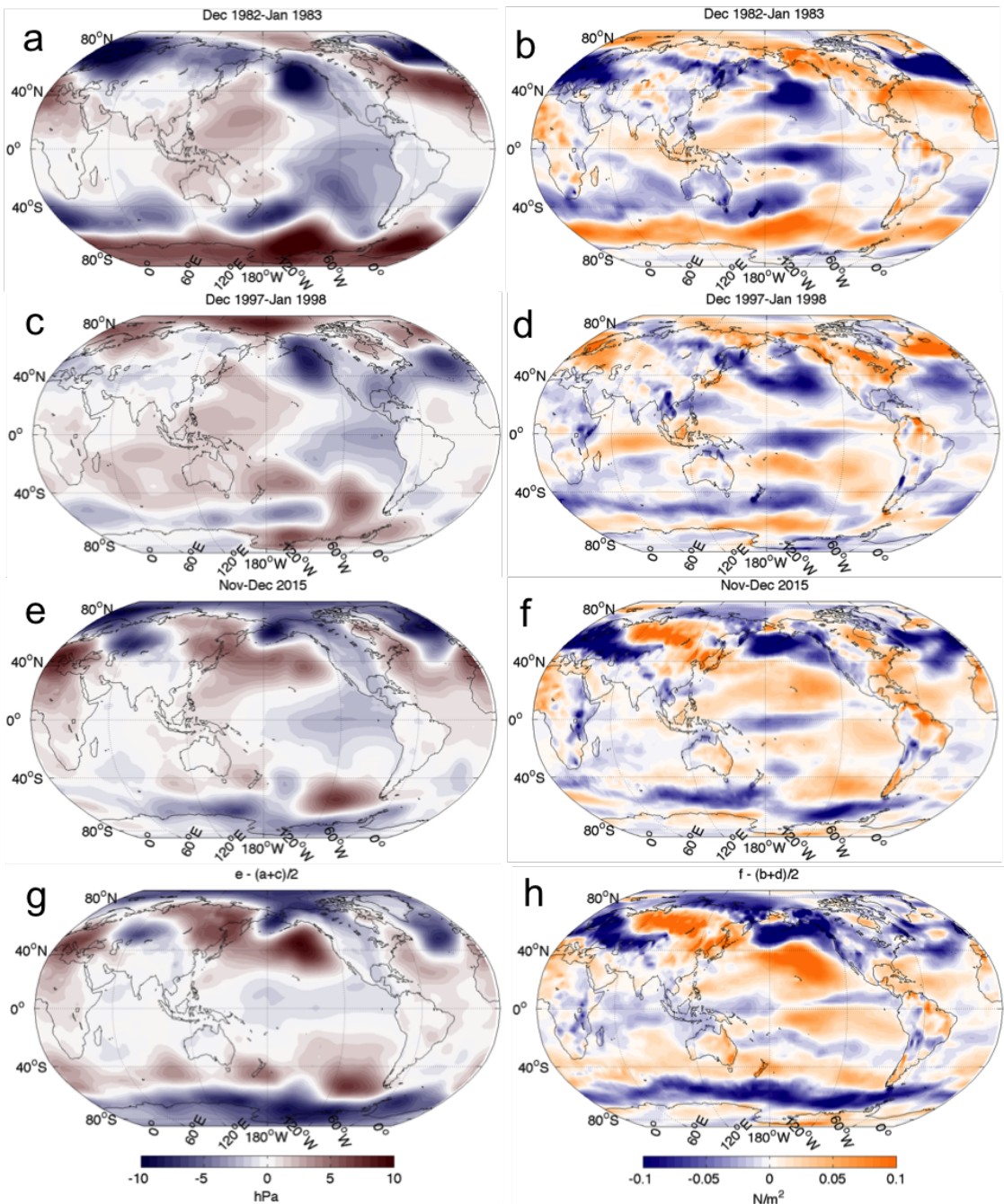

**Figure 3.** (Left; a, c, and e) The surface pressure and (Right; b, d, and f) zonal friction drag anomalies averaged over December-January for the 1982-83 and 1997-98 EP events, and averaged over November-December for the 2015-16 EP/CP event. The bottom-left (-right) map shows the differences in pressure (zonal friction drag) anomaly between the 2015-16 situation and the average of the 1982-83 and 1997-98 situations.

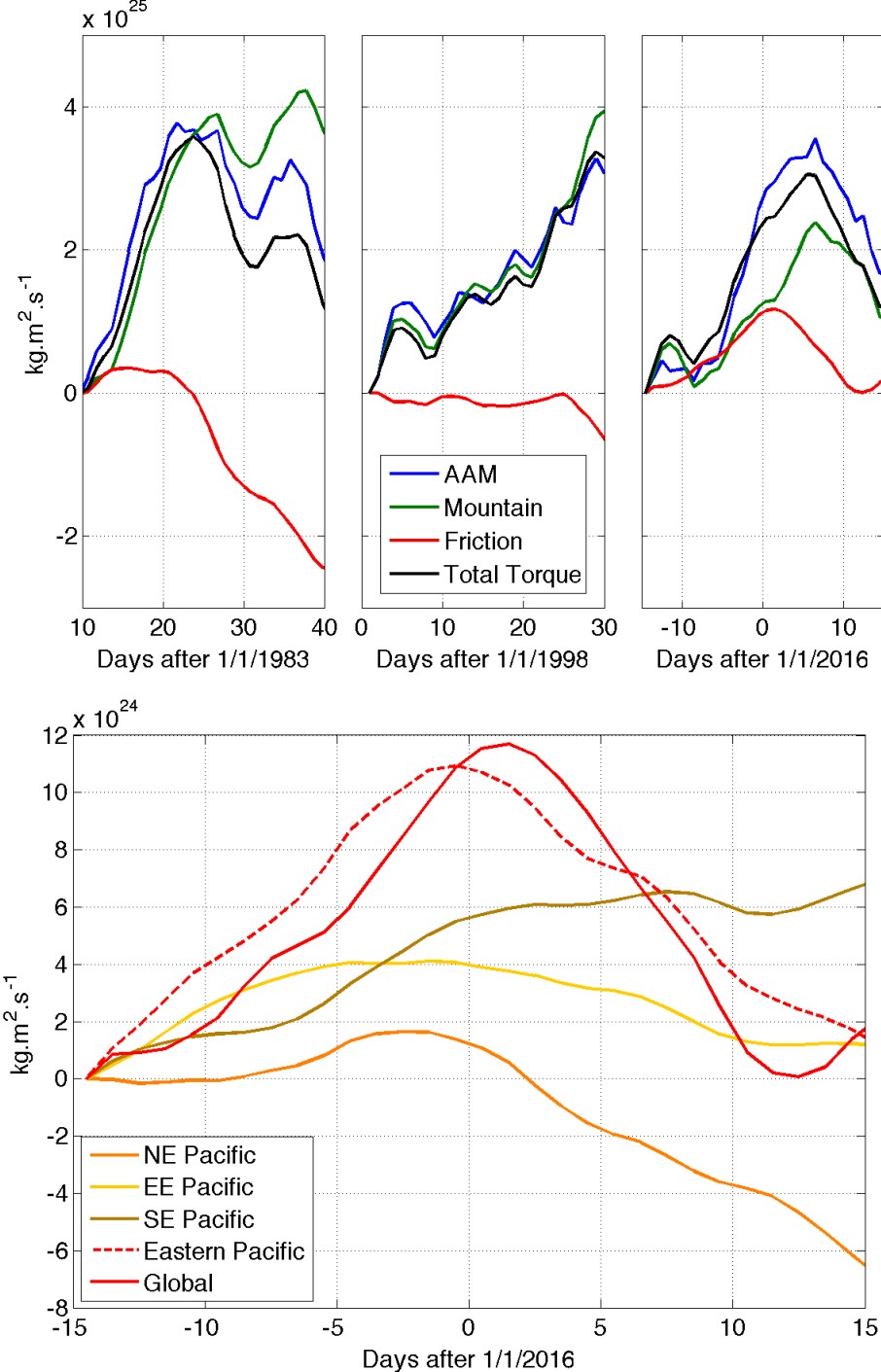

**Figure 4.** (Top) The integrated torques compared to AAM during the three events and (Bottom) the integrated friction torque with contribution from the different regions of the Pacific ocean during the 2015-16 event.

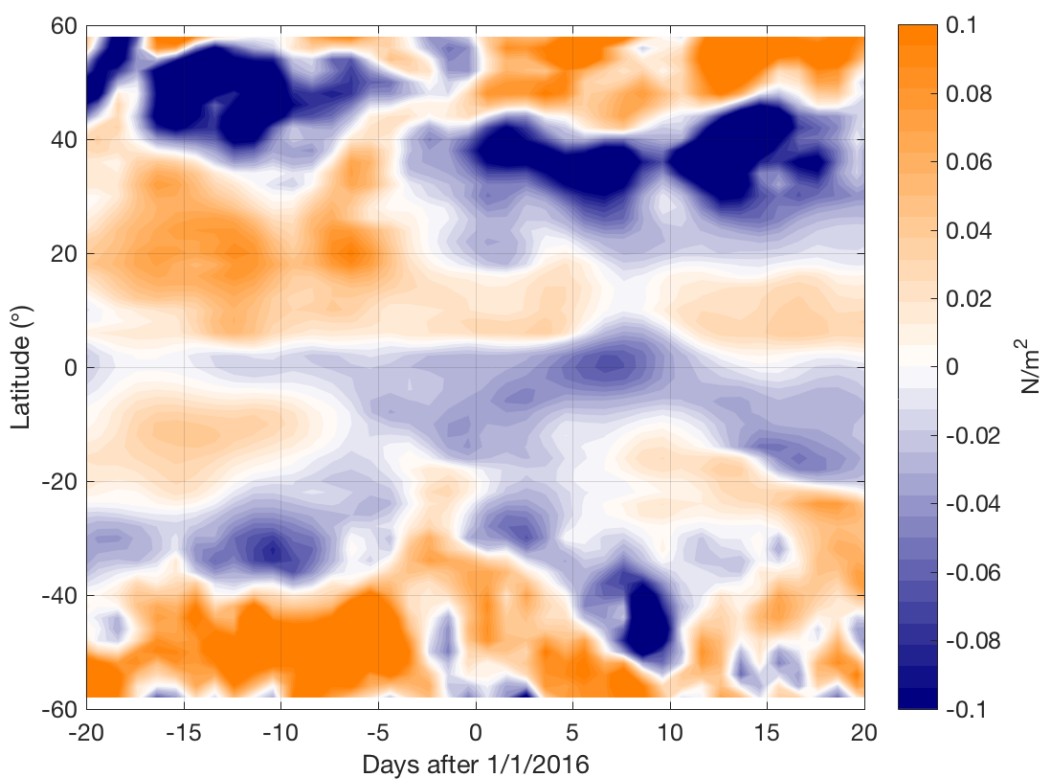

**Figure 5.** Time-latitude (Hovmoeller) diagram of the zonal friction drag anomaly between 11 December 2015 and 20 January 2016 and averaged over longitudes between $180°$ and $270°$.

**Table 1.** Contributions to the mountain and friction torques exerted by the solid Earth onto the atmosphere in Hadley (i.e., $10^{18}$ N.m), averaged over December-January for the 1982-83 and 1997-98 EP events (columns D82-J83 and D97-J98, respectively), and averaged over November-December for the 2015-16 EP/CP event (column ND15).

| | D82-J83 | D97-J98 | ND15 |
|---|---|---|---|
| Mountain Torque | | | |
| Global | 3.8 | 10.4 | −0.0 |
| Africa | 0.1 | 1.6 | −1.1 |
| Europe | −3.3 | 0.7 | −2.7 |
| Greenland | −3.8 | 2.1 | −2.5 |
| North America | 4.6 | 0.6 | −0.8 |
| South America | 6.6 | 6.0 | 6.1 |
| Asia | 1.8 | 1.2 | 3.3 |
| Oceania | −0.4 | −0.1 | 0.2 |
| Antarctica | 0.0 | −0.2 | −1.1 |
| Friction Torque | | | |
| Global | 0.7 | −2.0 | 4.6 |
| Africa | 2.8 | −0.5 | 1.9 |
| Europe | −2.9 | 1.2 | −2.6 |
| Greenland | −0.0 | 0.1 | −0.0 |
| North America | 2.3 | 2.0 | 0.7 |
| South America | 1.6 | 0.8 | 1.2 |
| Asia | −0.4 | −0.6 | 1.2 |
| Oceania | −0.7 | 0.2 | 0.2 |
| Antarctica | 0.1 | −0.1 | 0.1 |
| NE Pacific | −4.1 | −3.2 | 0.3 |
| NW Pacific | −0.3 | −1.0 | 0.8 |
| EE Pacific | −2.3 | −2.0 | 0.3 |
| EW Pacific | 1.4 | −0.0 | −0.3 |
| SE Pacific | 1.4 | 2.3 | 2.7 |
| SW Pacific | −0.7 | −0.4 | −0.4 |
| North Atlantic | 1.9 | −1.8 | −0.2 |
| Eq Atlantic | 0.5 | −0.2 | 0.2 |
| South Atlantic | −0.0 | −0.7 | −0.5 |
| Indian Ocean | −0.6 | 0.5 | −0.7 |
| Antarctic Ocean | 1.1 | 0.8 | −0.5 |