# Peer review of "Atmospheric Torques and Earth's Rotation: What Drove the Millisecond-Level Length-of-Day Response to the 2015-16 El Niño?"

_Earth System Dynamics, 2017_

## Referee Comment (RC1) · A. Speranza (Referee) · 16 Aug 2017

In this paper the largest El Niño/Southern Oscillation (ENSO) events of last forty years are investigated in their association with a significant (almost 1 ms) increase in the length of day (LOD).

Mountain and friction torques are estimated from available datasets and it is "found that: (i) as a mixed Eastern/Central Pacific event, the 2015-16 mountain torque was smaller than for the 1982-83 and 1997-98 events which were pure Eastern Pacific events, and (ii) the smaller mountain torque was compensated by positive friction torques arising from an enhanced Hadley-type circulation in the Eastern Pacific, leading to similar

[Figure]

AAM/LOD signatures for all three extreme ENSO events. The 2015-16 event thus contradicts the dominant paradigm that mountain torques cause the Earth rotation response for extreme El Niño events."

The presentation is neat and synthetic.

The addressed topic and the proposed results are interesting.

Just one remark: according to my own experience in estimating form-drag the sensitivity of the computed drag with respect to resolution of the adopted mountain representation can be relevant in areas where mountains are steep. The only mention I found in the paper of the mountain drag computation procedure is "For computation of the mountain torque, we used the model orography. The longitudinal gradients of the pressure field were computed with a five-point stencil.": more detail could help the reader. Was the dependence of the estimated drag on resolution checked?

Antonio Speranza

---

## Referee Comment (RC2) · Anonymous Referee #2 · 16 Aug 2017

This is a well-written paper but unfortunately one of limited scope and little science progress, with the analysis being essentially a repeat of Ponte & Rosen (1999) and de Viron & Dickey (2014, both studies cited in the manuscript). It is not clear in which way other disciplines or even the Earth rotation community will benefit from elaborations concerning the relative role of mountain and friction torques during different flavors of ENSO. For a non-linear phenomenon such as ENSO, it is not too surprising that the torques shape up differently for each event, and it will thus be difficult establishing any real 'paradigm'.

So does the finding of anomalous friction torques due to Hadley-type convection have

any implications for atmospheric modeling or our understanding of how ENSO evolves and to which physical processes it is connected? What will be the value added to measurements/predictions of Earth rotation? Will geodesists benefit from using atmospheric torques instead of AAM values or does this just lead to a loss of accuracy? Without addressing these points properly, it is difficult to see how the manuscript can be a sound contribution to a first-tier journal such as Earth System Dynamcis. The authors could turn to Ponte and Rosen (1999) for thoughts on the larger context, but even their discussion might not convince everybody as to why studies of atmospheric torques during ENSO events matter.

Some minor points:

- To strengthen the description of the characteristic torque patterns (presently time-invariant snapshots in Figure 3), would it be possible to plot the space-time evolution of the mountain/friction torques in the Pacific using some kind of Hovmoeller diagram? This might support the plain textual description of the various flows and circulation anomalies in Section 3 (page 5).

- The following recent review of the 2015/2016 ENSO should be cited in the manuscript: L'Heureux et al. (2017) Observing and Predicting the 2015/16 El Nino. BAMS vol. 98(7): 1363–1382.

- After Eq. (6), $\overline{LOD}$ is referred to as "conventional mean LOD". Do you mean the nominal length of the solar day? If so, update the formulation.
* * *

---

## Author Comment (AC1) · 5 Sep 2017

The orography is certainly an important parameter for computing the mountain torque, thus taking, or not, the best mountain representation as possible is a key issue.

Nevertheless, the pressure, wind, and zonal friction drag data on which we based our torque and AAM computation were obtained with the original ERA-Interim model orography. Using another orography for computing the torque/AAM would mean taking the risk that, e.g., the pressure data is not consistent with the orography, and therefore propagating biases and errors into the torque, which would eventually result in a non-closed AAM budget. By choosing to take the orography model distributed by

the ECMWF together with their atmospheric data, we guarantee that the AAM and torques we computed are consistent. Furthermore, as noted in lines 30-32 of p. 5 in the manuscript, torques related to sub-grid scale processes are generally considered to be small at these time scales (e.g., de Viron et al., 1999).

How did we check that our torque and AAM were consistent and somehow 'realistic'?

(1) We checked the closure of the AAM budget: the time derivative of the AAM should be equal to the torque. This item was verified to less half a Hadley for the 1997-98 and 2015-16 winters, and about 1.3 Hadley for the 1982-83 winter, using the ERA ECMWF data. Interestingly, this is far from being the case using NCEP/NCAR Reanalysis data for a reason that one should investigate but remains out of the scope of this paper (see Fig. 1 below).

(2) The angular momentum and the time-integrated torque should be in agreement with the observed LOD. This is the case to a reasonable extent that allows us to reach the conclusions of our article. It means that the combination of the ERA data and the 'ERA orography' creates AAM anomalies of the right magnitude and at the right time that explains the observed LOD, as shown by the near-agreement of the blue and black curves in Fig. 4 (top) of the manuscript.
* * *
**Winter AAM Budget**

Legend: ERA (magenta), NCEP (green)

X-axis: ENSO Year (1983, 1998, 2016)
Y-axis: Hadley (-1 to 7)

**Fig. 1.** The difference between dAAM/dt and the mountain+friction torque averaged over the ENSO winter 1982-83, 1997-98, and 2015-16.

---

## Author Comment (AC2) · 5 Sep 2017

Following the reviewer suggestion, we worked on improving the quality, the readability, and the scope of our paper in the new version of the manuscript. The manuscript was modified in consequence.

It is always difficult to reply to an evaluation about the interest or not of a paper, as part of it is a matter of point of view. We do consider, of course, that our work is new enough to deserve publication in a top-level journal such as ESD, as it bring new insight on how Earth and atmosphere exchange angular momentum during an ENSO event or, to be more specific, it shows that this exchange can differ to a considerable extent

from one event to the other. Basically, there exist only three extreme ENSO events in recent history - including the 2015-16 one. This number is so small that any new event is worth being studied from several points of view, among which is the Earth rotation view point we adopt.

Shortly after (and even during) this event, researchers (e.g., M. L'Heureux at 2016 EGU, and more recently Paek et al. 2017 in a GRL paper) agreed on the fact that the 2015-16 event was of a nature different from its extreme predecessors. It was basically the first time that one could observe a strong, mixed EP/CP event, and therefore a unique chance to see if the impressive observed rotational signal in the LOD was produced by the same processes as in the pure EP events of 1982-83 and 1997-98.

As we wanted to study the phenomenon from the interaction point of view, we naturally used the torque approach, that was already used in other studies of Earth-Atmosphere interaction in the context of ENSO events, i.e., those of R. Ponte, R. Rosen, J. Dickey, and O. de Viron. Using this approach does not mean that we repeat older studies, but this is the approach that we can apply to better understand the AM exchanges, and better explain the differences between the older events and the latter one.

Is the new story the same as in these previous studies? R. Ponte and R. Rosen showed that coherent mountain torques over the Himalayas, the Andes, and the Rockies created the LOD anomaly during the 1982-83 ENSO, while O. de Viron, S. Marcus, and J. Dickey showed that for the strong 1988-89 La Niña, it was exactly the opposite, consistent with the relation between the Niña and Niño. In de Viron and Dickey (2014), the authors showed that the same conclusion can be drawn using all the ENSO event, and not only the strong events, and that CP events generated a weaker mountain torque. Here, we show that the 2015-16 event, which was, as pointed out by various researchers, of a different nature, resulted in a similar impact on the LOD, but with very different Earth-atmosphere interaction processes, where the mountain torque alone cannot explain the LOD anomaly, and where the friction torque - that often simply damps the anomaly created by the mountain torque - gives a significant positive contribution, driving the LOD anomaly to the same near-millisecond values of the previous ENSO events. Therefore, we show that the observed LOD anomaly was not created by the mountain part of the torque - as one could have thought in virtue of the previous studies (what we now refer to as the 'existing paradigm' - see for example the second sentence in Section 4 of de Viron and Dickey 2014) - but by a mixture of the mountain and friction torques. We think this fact is worth being pointed out, as it relates directly to the changed Earth system dynamics that gave rise to the pronounced rotational signature of this new type of event.

We add a new piece of text to Section 2 of our paper, pointing out that while the angular momentum approach is most suitable for the now-casting or forecasting of LOD, the torque approach can provide dynamical insight into the mechanisms generating the rotational anomalies, and can also serve as an internal consistency check for the models under the extreme conditions accompanying these events. While the strength of frictional coupling (i.e., momentum transfer or torques) between the atmosphere and thermocline layer is a key to the dynamical evolution of ENSO events (e.g., Neelin 1998), this topic is beyond the scope of our paper with its focus on Earth rotation. For the interest of the broader ESD community, however, a new paragraph has been added to the final section of the paper, discussing how these changing atmospheric torques might interact with recent climatic trends.

We also add a new figure - as supplemental material (see the attached file) - showing the time-latitude (Hovmoeller) diagram of the zonal friction drag anomaly in the Eastern Pacific and the evolution of the integrated (positive) friction torque. We completed the explanation by additional text in the last paragraph of Section 3.

The other two minor points were also addressed: the value of the standard LOD (of 86400 s) was referred to as the 'nominal LOD'; we added a reference to L'Heureux et al. (2017) in the text (first sentence of the second paragraph of Section 4).

Please also note the supplement to this comment:

https://www.earth-syst-dynam-discuss.net/esd-2017-52/esd-2017-52-AC2-supplement.pdf

[Figure]

**Supplement:**

**Supplementary Material**

[Figure]

**Figure S1:** Left: Time-longitude (Hovmoeller) diagram of the zonal friction drag anomaly between 15 December 2015 and 15 January 2016 and averaged over latitudes between 60°S and 60°N. Right: Reproduction of the integrated friction torque arising from the Eastern Pacific zone as shown by the red, dashed line in the bottom panel of Fig. 4.

---

## Author Response (AR1)

**Response to the Editor**

September 27, 2017

Dear Editor,

Following the reviewers' suggestions, we worked on improving the quality, the readability, and the scope of our paper in the new version of the manuscript. The manuscript was modified in consequence as explained in the two responses to the reviewers. The main modifications are made in bold characters.

As you suggested in your last report, we added a paragraph (lines 20-25, page 3) to clarify the question of the torque sensitivity to the resolution of the orography grid raised by reviewer #1, thereby explaining why we used the ERA orography grid, and taking advantage of a recently published paper on the topic. We reproduce here this additional paragraph:

*For computation of the mountain torque, we used the model orography at its native $2° \times 2°$ resolution, thereby ensuring consistency between the wind, pressure, and zonal momentum flux data sets and the derived AAM and torque quantities. A recent study by van Niekerk et al. (2016) found that resolved mountain torques in the Met Office United Model with free atmospheric wind and temperature relaxed to ERA-Interim reanalyses are relatively insensitive to increasing model resolution (see, e.g., their Fig. 7), although they are more strongly impacted by large-scale (synoptic) processes than are the parameterized sub-grid scale torques (not considered in our study).*

With the last revision round, we proposed to add a supplemental material file with a figure (referred to as Fig. S1). That figure was a time-longitude (Hovmoeller) plot of the zonal friction drag anomaly in the Eastern Pacific during the last event. We actually replaced Fig. S1 by a time-latitude plot of the same quantity over the same span that we found showing more explicitly the time and space locations of the friction 'bursts'. The corresponding documenting paragraph in the text is as reproduced below:

*A Hovmoeller (time-latitude) plot of the Eastern Pacific frictional drag contributing to the 2015-16 LOD maximum (Fig. S1) highlights its three-belt structure, and shows the EE Pacific contribution (spanning $15°N$-$15°S$) to arise from two areas: one in the southern hemisphere originating from inflow to the westward-displaced boreal winter Hadley circulation, and one in the northern hemisphere originating from enhanced easterly flow on the*

*equatorward flank of the NE Pacific high pressure area (similar to the Nov-Dec 2015 pressure anomalies seen in Fig. 3e).*

Best regards,

S. B. Lambert, S. L. Marcus, O. de Viron

---

## Author Response (AR2)

**Response to the Editor**

October 11, 2017

Dear Editor,

We addressed the two points raised in your report:

*1) you should explain the derivation of Eq. 6 and in particular the meaning of the surface loading deformation (for which there is no reference).*

We added the following explanation (in bold in the manuscript) and references:
"the mass term is evaluated using the inverted barometer assumption (Jeffreys, 1916) to account for the quasi-static response of the oceans to atmospheric pressure loading, and the factor of 0.7 accounts for the compensating changes in the moment of inertia arising from the elastic deformation of the solid Earth in response to the surface loading (Munk and MacDonald, 1960; Barnes et al., 1983)."

2) you should include figure S1 in the main manuscript and not as supplementary material.

Figure S1 is now inserted in the manuscript as Fig. 5 and the relevant modifications have been made in the text.

Best regards,

S. B. Lambert, S. L. Marcus, O. de Viron